# The Failure of Molecular Imprinting in Conducting Polymers: A Case Study of Imprinting Picric Acid on Polycarbazole

Karolina Głosz [1], Magdalena Fabin [1], Patryk Janasik [1], Weronika Kołodziej [2,†], Agnieszka Stolarczyk [1,*] and Tomasz Jarosz [1,*]

[1]   Department of Physical Chemistry and Technology of Polymers, Silesian University of Technology, 44-100 Gliwice, Poland; karolina.glosz@polsl.pl (K.G.); magdalena.fabin@polsl.pl (M.F.)
[2]   Faculty of Chemistry, Silesian University of Technology, 44-100 Gliwice, Poland; werokol490@student.polsl.pl
*   Correspondence: agnieszka.stolarczyk@polsl.pl (A.S.); tomasz.jarosz@opayq.com (T.J.)
†   Student at the Faculty of Chemistry.

**Abstract:** The aims of this study were to investigate the potential of utilising molecularly imprinted polycarbazole layers to detect highly toxic picric acid (PA) and to provide information about their performance. Quantum chemical calculations showed that strong interactions occur between PA and carbazole (bond energy of approximately 31 kJ/mol), consistent with the theoretical requirements for effective molecular imprinting. The performance of the sensors, however, was found to be highly limited, with the observed imprinting factor values for polycarbazole (PCz) layers being 1.77 and 0.95 for layers deposited on Pt and glassy carbon (GC) electrodes, respectively. Moreover, the molecularly imprinted polymer (MIP) layers showed worse performance than unmodified Pt or GC electrodes, for which the lowest limit of detection (LOD) values were determined (LOD values of 0.09 mM and 0.26 mM, respectively, for bare Pt and MIP PCz/Pt, as well as values of 0.11 mM and 0.57 mM for bare GC and MIP PCz/GC). The MIP layers also showed limited selectivity and susceptibility to interfering agents. An initial hypothesis on the reasons for such performance was postulated based on the common properties of conjugated polymers.

**Keywords:** molecularly imprinted polymer; sensor; picric acid; 2,4,6-trinitrophenol; detection; polycarbazole

## 1. Introduction

In recent years, the increasing scientific interest in the subject of environmental protection and safety has led to tremendous scientific developments in the detection of hazardous substances such as explosives [1,2]. Energetic nitroaromatic compounds, which include 2,4,6-trinitrophenol (PA), have attracted particular attention [3–5]. One common application of PA is the use of the compound as a standard material for analytical methods such as HPLC [6]. Due to its toxic and carcinogenic properties, it is extremely important that PA can be detected even in trace amounts [7,8]. Picric acid may cause damage to the eyes and skin, anemia, liver injury, and respiratory system damage [7,9]. For male and female F344 rats, the $LD_{50}$ doses for oral administration of PA were determined to be 290 and 200 mg/kg respectively [10]. It has been reported that ingestion of 1 to 2 g of picric acid causes severe poisoning in humans [11].

Due to the relatively high solubility of picric acid in water, even the smallest concentration of PA in water is intolerable. Maximum permissible concentrations have been established for this compound, e.g., by the National Institute for Occupational Safety and Health (NIOSH) and the Occupational Safety and Health Administration (OSHA)—according to the TWA method, the contamination of maximum PA in the air should not exceed 0.1 mg/m³ [12].

The most commonly used methods for detecting PA include mass spectrometry [13], the use of field-effect transistors [14], and fluorescence spectroscopy [15]. Unfortunately,

the problem with the above methods is the structure of the compound. Due to the similarity of the chemical structure of PA to the structures of other nitroaromatic compounds, it is difficult to differentiate between them and PA when using, for example, photo-induced electron transfer, resonance fluorescence energy transfer, or strong electrostatic interactions [16]. These methods are, therefore, limited by their poor selectivity or complicated procedures. Therefore, it is very important to develop a highly sensitive method for the detection of PA.

Many examples of selective sensors for the detection of PA and other nitroaromatic compounds have already been described in literature. These sensors are often based on metal–organic frameworks [17,18] or carbon dots [19], even if other materials are also applied for this purpose. The disadvantages of these sensors are the complex manufacturing process and their high unit costs [20].

Molecular imprinting is a group of processing methods aimed at producing layers containing pores, whose shape and size match that of a selected template molecule. Typically, molecularly imprinted polymers (MIPs) are produced via the polymerisation of an adduct between the template molecule and a monomer. The resultant polymers, after the removal of the template molecule from the polymer matrix, allow for the specific adsorption of that template, which is of great significance for producing highly selective sensors. Particular research attention has been given to molecularly imprinted conjugated polymers, such as polycarbazole or polypyrrole [21–23].

Efforts have also been undertaken to utilise MIPs for the detection of PA (Table 1). Despite the existence of a few reports, no data about the effect of imprinting or comparisons with bare electrodes have been provided, making it impossible to identify the effect of molecular imprinting on the detection parameters of these sensors.

**Table 1.** Example of MIP sensors reported for the detection of PA.

| MIP Receptor Layer | Media | LOD | Ref. |
|---|---|---|---|
| MIP/rGO/PGE [a] | Water and soil | 1.4 µmol/L | [24] |
| BTAM [b] | Acetonitrle-to-toluene (95:5) | 0.2 ng/L | [25] |
| N-CDs@MIP [c] | Water | 0.15 nM | [26] |

[a] Pyrrole (MIP), reduced graphene-oxide-coated pencil graphite electrode. [b] Bis(2,2'-bithienyl)-(4-aminophenyl)methane. [c] Bitrogen-passivated carbon dots infused with a molecularly imprinted polymer (3-aminopropyltriethoxysilane).

To obtain selective (with a high response only to the intended analyte and preferably no response to other analytes) [27] and sensitive (low LOD) [28] MIPs, it is important to examine if the template is compatible with the monomer (i.e., if there are interactions between them) [29]. The most common technique used to produce MIPs is the self-assembly approach, followed by the polymerisation of the monomer, which relies on non-covalent interactions, e.g., hydrogen bonds [30], ionic/hydrophobic interactions, etc. [31]. The advantage of this type of interaction is the easy removal of the template from the template–monomer complex, e.g., extraction with a solvent [32] or immersion in a solvent [33]. Due to the fact that non-covalent interactions are easily disrupted, it is important to choose a monomer–template pair that will create complex with strong interactions between them [34]. It has been confirmed that higher-energy bonding leads to the formation of an adduct with stronger interactions, resulting in a more selective MIP [35].

In this work, we have provided theoretical background for the interactions between picric acid (PA) and a conjugated monomer, i.e., carbazole, based on quantum-mechanical calculations. We investigated the process of producing a MIP polycarbazole layer on platinum and glassy carbon electrodes and investigated their performance in detecting PA.

## 2. Materials and Methods

The following reagents were used in this work: acetylsalicylic acid (>99%, Sigma-Aldrich, St. Louis, MO, USA), sulfuric acid (>95%, Chempur, Karlsruhe, Germany), potassium nitrate (>95%, POCH S.A, Gliwice, Poland), carbazole (>97%, TCI, Tokyo, Japan), and tetrabutylammonium tetrafluoroborate ($Bu_4NBF_4$) (>98%, TCI).

### 2.1. Synthesis of 2,4,6-Trinitrophenol

Sulphuric acid (60 mL, 1.12 mol) was introduced into a three-necked flask equipped with a mechanical stirrer. Next, acetylsalicylic acid (6 g, 0.03 mol) was added in small portions over the course of approximately 60 min. After the addition of acetylsalicylic acid, the mixture was heated for 60 min at 115–120 °C. Next, the reaction mixture was cooled to approximately 70 °C, and potassium nitrate (13.5 g, 0.134 mol) was introduced in small portions, resulting in the temperature rising to 80–95 °C and being kept in that range. After all of the potassium nitrate had been added, the reaction mixture was heated up to 120 °C and stirred for 20 min. Following this, the heating was disengaged. After the mixture had cooled to room temperature, the contents of the flask were transferred to a tall beaker of deionised ice water. The precipitate was filtered under a vacuum and rinsed twice with small amounts of deionised water. Next, the raw product was recrystallized from deionised water. After the mixture cooled, the precipitate was filtered off and dried, resulting in 2,4,6-trinitrophenol (4.71 g, 0.021 mol). A summary of the reaction is presented in Figure 1. The yield of the reaction was 70%. PA melting point: 122.5 °C (capillary method), $^1$H NMR (300 MHz, DMSO-$d_6$) $\delta$ (ppm): 8.59 (s, 2H, Ar-H). IR-ATR (diamond) (Figure A2): 3108 cm$^{-1}$ $\nu$ (O-H) 2870 cm$^{-1}$ $\nu$s (C-H), 1630 cm$^{-1}$ $\nu$ as ($NO_2$), 1606 cm$^{-1}$ $\nu$ (C=CAr), 1341 cm$^{-1}$ $\nu$s (C-N), 1275 cm$^{-1}$ $\nu$ (C-O), 1154 cm$^{-1}$ $\nu$ (C-H) in-plane bending, 779 cm$^{-1}$ $\nu$ (C-$NO_2$), 703 cm$^{-1}$ $\nu$ (C-H out-of-plane bending, 663 cm$^{-1}$ $\nu$ (C-$NO_2$ wagging. Raman spectroscopy (laser 840 nm) (Figure A3) : 1636 cm$^{-1}$ C-C ring str., 1348 cm$^{-1}$ $\nu$ $NO_2$asym, 1280 cm$^{-1}$ $\nu$ $NO_2$sym, $\nu$ C-N str, 831 cm$^{-1}$ $\sigma$ $NO_2$ in plane (scissoring).

**Figure 1.** Schematic representation of the synthesis of PA.

### 2.2. Electrochemical Investigations

Molecularly imprinted polymer (MIP) and non-imprinted polymer (NIP) layers were produced via electrochemical polymerisation. Electrochemical polymerisation was conducted using cyclic voltammetry in acetonitrile solutions containing 0.1 M tetrabutylammonium tetrafluoroborate ($Bu_4NBF_4$/MeCN) as a supporting electrolyte and 20 mM carbazole as the monomer. NIP films were produced directly from this solution, whereas the MIP layers were produced from solutions supplemented with 80 mM PA.

For electrochemical polymerisation, constant-surface-area electrodes made out of either platinum or glassy carbon were utilised as working electrodes. A platinum coil was used as the counter-electrode, and silver wire was used as the pseudoreference electrode. In the cases of both NIP and MIP layers, the parameters of the cyclic voltammetry experiments were identical and were as follows: the applied working electrode potential range was −0.5 V to +1.85 V, the potential scan rate was 0.1 V/s, and 10 potential cycles were conducted.

The synthesized MIP and NIP layers were investigated in PA solutions of varied concentrations via differential pulse voltammetry (DPV). The initial potential in DPV was 0.2 V, and the final potential was −2 V. The step potential was −0.005 V, the modulation amplitude was −0.035 V, the modulation time was 0.05 s, and the interval time was 0.5 s. The electrode setup that was utilised was identical to that described above for the electrochemical polymerisation experiments.

The imprinting factor (IF) was calculated as the ratio of the peak current observed for the MIP layer to the peak current observed for the NIP layer. The IF values were calculated for layers deposited on Pt that were used to detect PA, as well as the two selected interfering agents. The IF values were also determined for PCz layers deposited on the GC electrodes used for the detection of PA.

For the purpose of conducting cross-selectivity investigations, nitrobenzene (9 mM) and nitromethane (18 mM) were used as interfering agents. The cross-selectivity was investigated via DPV by utilising the same experimental parameters as in the case of the measurements conducted for the detection of PA.

### 2.3. Quantum Chemical Calculations

For the calculations, DFT/TDDFT (Time-Dependent Density Functional Theory) was used with the B3LYP [36] hybrid functional combined with the 3–21 G(d) basis set. For all optimised structures, the frequency calculations were systematically achieved (at the same level of theory) to confirm the minimum nature of the optimised geometries. All calculations in this work were performed using the ORCA 4.1.1 [37] package programs. Input files and molecular orbital plots were prepared with the Gabedit 2.4.7 software [38].

### 2.4. SEM Analyses

The morphology of MIP PCz and NIP PCz layers deposited on the Pt electrodes was investigated using a Phenom ProX (Waltham, MA, USA) scanning electron microscope (SEM). The basic SEM operation parameters were the following: The working distance was 10–11 mm, the acceleration voltages of the incident electron were 15 kV, and images were recorded at a 6000× and 15,000× magnification.

## 3. Results and Discussion

### 3.1. Investigation of Polymer Layers Deposited on Platinum Electrodes

SEM investigations revealed that the inclusion of PA as the template molecule during polymerisation had a slight effect on the overall morphology of the polymer films produced via electrochemical polymerisation (Figure 2). More relevantly, the polymerisation of carbazole in the presence of PA resulted in a lower degree of coating on the electrodes than in the case of polymerisation conducted without the presence of PA. This is indicative of the electrochemical polymerisation being hampered due to interactions that stabilised the monomer and hindered its oxidation. This was in line with the results of the quantum chemical modeling, which predicted the formation of a hydrogen bond between carbazole and picric acid.

Regardless of whether a modified or unmodified electrode was used, the voltammograms of the PA solutions contained current signals corresponding to the electrochemical reduction of PA. Due to the fact that acetonitrile is a polar aprotic solvent, this reduction was expected to take place with the transformation of the nitro functionality not into an anionic species, expected based on literature [39], but into a hydroxylamine functionality due to the self-protonation of picric acid originating from its highly acidic phenol hydroxyl group [40].

In the case of the unmodified Pt electrodes (Figure 3A), three distinct electron transfer stages were observable at −0.38 V, −0.64 V, and −0.94 V, respectively, corresponding to the irreversible reduction of each nitro group, which was corroborated by cyclic voltammetry (Figure A1).

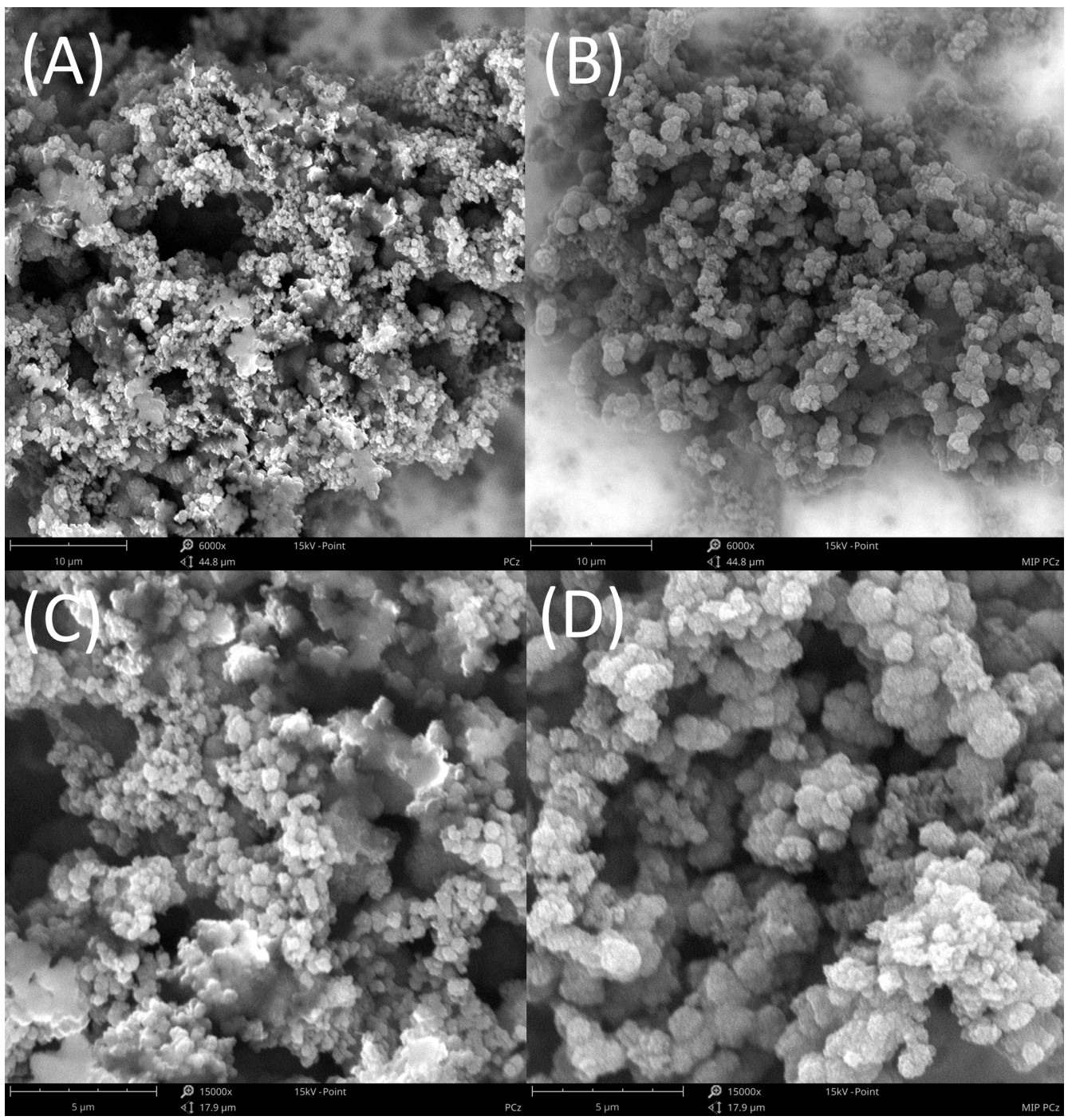

**Figure 2.** SEM images of NIP PCZ/Pt (**A**,**C**) and MIP PCz/Pt electrodes (**B**,**D**); magnification: (**A**,**B**) 6000×; (**C**,**D**) 15,000×.

DPV measurements, conducted for a range of PA concentrations (Figure 3), evidenced the expected decrease in the peak current with decreasing PA concentration. In the case of PA at a concentration of 0.10 mM, a shift in the observed peak potential towards more negative potentials (Figure 3C) and towards more positive potentials (Figure 3A) was observed. In the case of the unmodified Pt electrode, this was likely due to the occurrence of the specific adsorption of PA, resulting in most of the active sites on the electrode being occupied above a certain PA concentration in the working solution. Once a sufficiently low PA concentration was employed, the number of adsorbed PA molecules no longer exhausted the number of active sites on the electrode, leading to a decrease in the "overpotential" caused by the saturation of active sites.

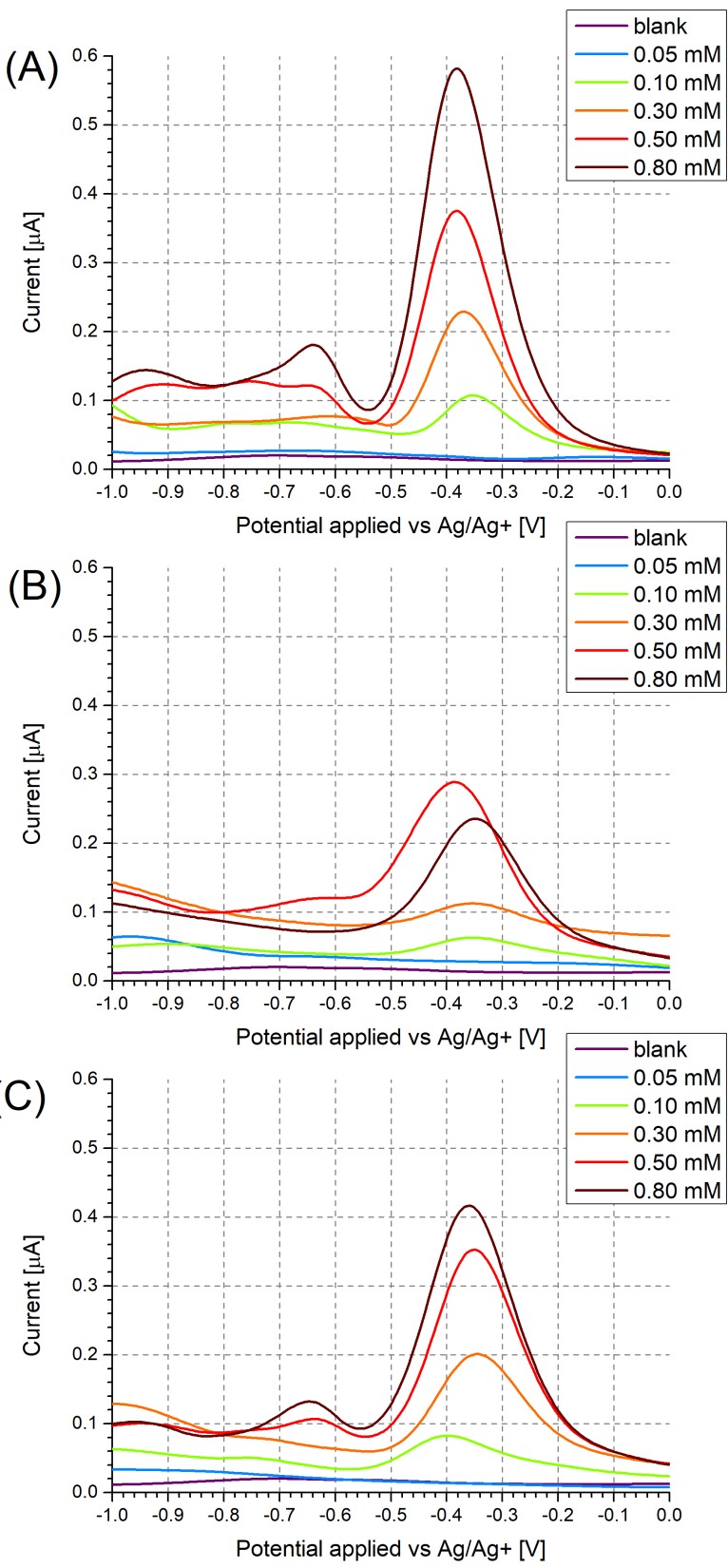

**Figure 3.** Differential pulse voltammograms recorded for (**A**) unmodified Pt electrodes, (**B**) NIP PCz/Pt electrodes, and (**C**) MIP PCz/Pt electrodes. The voltammograms were recorded in $Bu_4NBF_4$/MeCN solutions containing 0.05–0.80 mM PA.

In the case of the MIP PCz/Pt electrode, the strong interactions between PA and carbazole are likely to hinder the desorption of even the reduced form of PA from the polymer surface. Due to this, at a sufficiently low concentration of PA in the working solution, all electrode active sites may become occupied by the reduced form of PA, which undergoes further reduction at more negative potentials, contributing to the observed shift in the peak potential.

Regardless of the choice of electrode and its modification, the reduction peaks are no longer observed for PA concentrations of 0.05 mM. Due to this, solutions containing even lower PA concentrations were not investigated.

Based on the recorded DPV results, the NIP PCz/Pt electrode showed the least sensitivity to PA (Figure 3B). Moreover, the peak current observed when using this electrode is not proportional to the changing PA concentration, as a higher current is repeatedly observed for solutions containing 0.50 mM PA than for solutions containing 0.80 mM PA. The performance of each system can also be evaluated by using a model of the data based on, e.g., a logarithmic dependence in the form of $y = b \cdot \ln(x - a)$ (Figure 4). In the case of the NIP PCz/Pt electrode (Figure 4B), the b factor in the modeled equation, which translates to the scale of the response of the electrode to a unit PA concentration, is the least favourable and equals $3.8 \times 10^{-7}$. The highest sensitivity to PA was observed in the case of the unmodified Pt electrodes (Figure 4A), with a b factor equal to $9.8 \times 10^{-7}$, with the performance of the MIP PCz/Pt electrode exhibiting average performance, contrary to the expectations.

This trend is mirrored in the limit of the detection values calculated for each of the investigated electrodes (Table 2). It should be noted that in the case of Pt, molecular imprinting resulted in an approximately threefold reduction in the LOD in comparison with the NIP PCz/Pt electrode. However, due to the fact that platinum is an effective catalyst of redox reactions, far more so than polycarbazole, the lowest LOD value was observed in the case of unmodified Pt electrodes.

**Table 2.** Summary of the calculated values of the limit of detection for PA and the b factors in the modeled dependence for each of the investigated electrodes.

| Electrode | b Factor [a] | Limit of Detection (LOD) [b] |
|---|---|---|
| Unmodified Pt | $9.8 \times 10^{-7}$ | 0.09 mM |
| NIP PCz/Pt | $3.8 \times 10^{-7}$ | 0.62 mM |
| MIP PCz/Pt | $7.6 \times 10^{-7}$ | 0.26 mM |
| Unmodified GC | $5.4 \times 10^{-7}$ | 0.11 mM |
| NIP PCz/GC | $1.4 \times 10^{-7}$ | 0.12 mM |
| MIP PCz/GC | $1.4 \times 10^{-7}$ | 0.57 mM |

[a] Calculated based on modeling the experimental data using the following function: $y = b \cdot \ln(x - a)$. [b] Calculated from LOD = $3\sigma/s$, where $\sigma$ is the standard error of the estimate and s is the slope of the curve.

### 3.2. Investigation of Polymer Layers Deposited on Glassy Carbon Electrodes

DPV measurements were also conducted using modified and unmodified GC electrodes (Figure 5). Contrary to what was observed for Pt electrodes, no significant potential shift of the current peaks was observed. Although specific adsorption is also expected to take place on the GC electrode, it will occur on a much smaller scale than in the case of platinum. In the case of the MIP PCz/GC electrode, the potential shift caused by interactions between PA and carbazole is only marginal due to the incomplete coverage of the electrode with the polymer layer.

In the case of NIP PCz/GC electrodes, two peaks are observed (Figure 5B) at potentials corresponding to the peaks observed for unmodified GC electrodes (Figure 5A) and for MIP PCz/GC electrodes (Figure 5C), respectively. This is attributed to the NIP PCz/GC electrodes having only partial coverage of the polymer film, possibly due to the degradation of the polymer film during the prolonged electrochemical polymerisation. In this case, PA

will interact and undergo electrochemical reduction simultaneously in areas where the GC electrode was exposed and on the surface of the polymer layer.

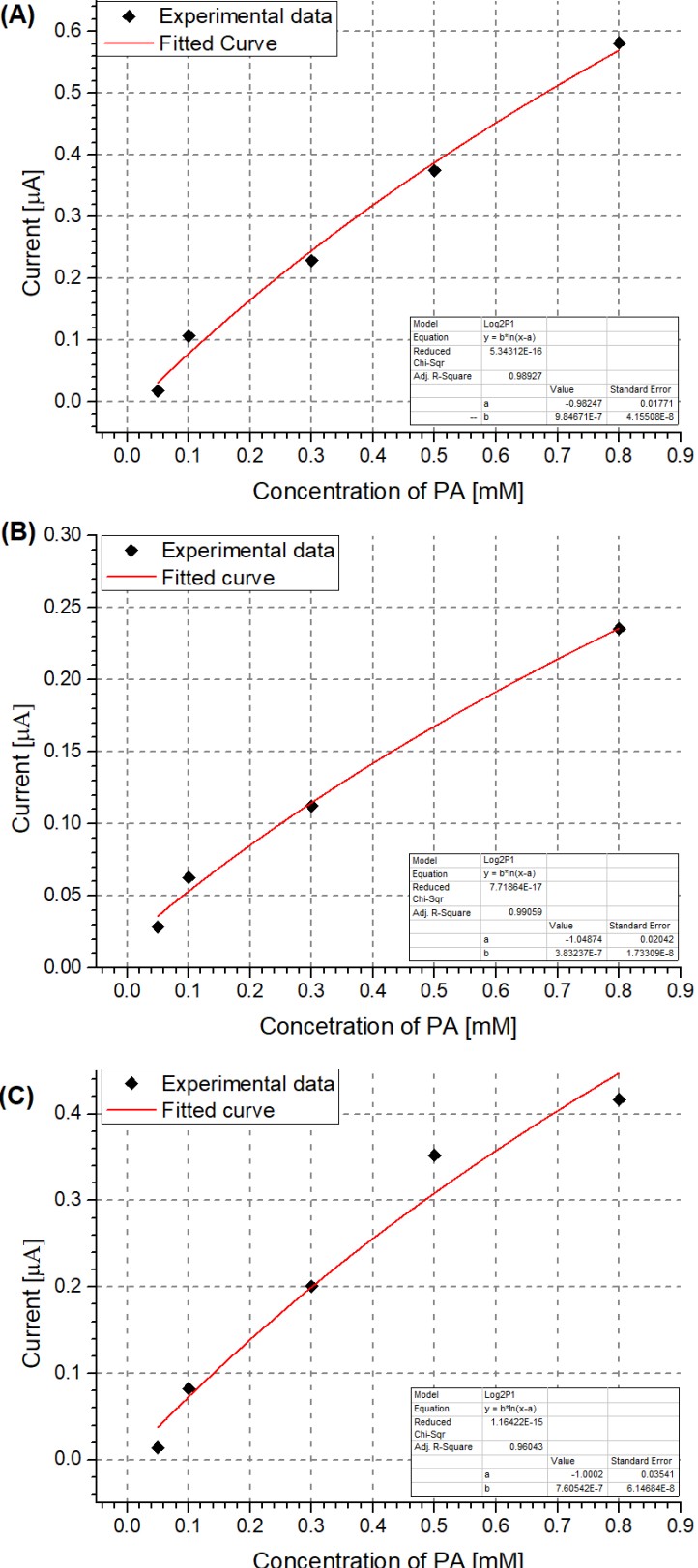

**Figure 4.** Calibration curves for the detection of PA using the investigated electrodes: (**A**) unmodified Pt electrode; (**B**) NIP PCz/Pt electrode; (**C**) MIP PCz/Pt electrode.

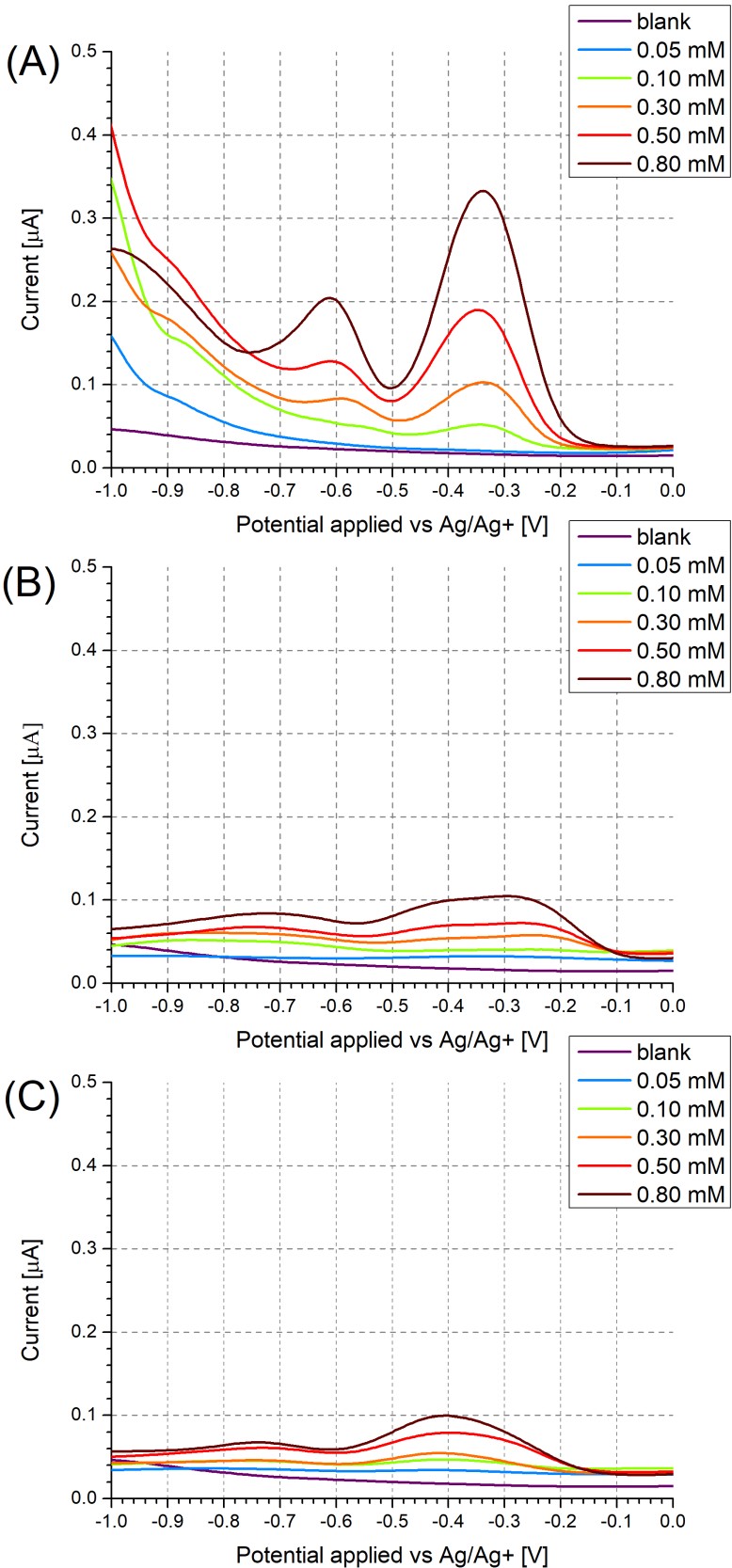

**Figure 5.** Differential pulse voltammograms recorded for (**A**) unmodified GC electrodes, (**B**) NIP PCz/GC electrodes, and (**C**)MIP PCz/GC electrodes. The voltammograms were recorded in $Bu_4NBF_4$/MeCN solutions containing 0.05–0.80 mM PA.

Similarly to the case of the Pt electrodes, the unmodified GC electrode shows the highest sensitivity to PA, as seen by its comparatively lowest LOD and best b factor (Table 2). Despite its b factor value being similar to that of the NIP PCz/GC electrode (Figure 6), the MIP PCz/GC electrode shows the lowest LOD among the three GC-based electrodes.

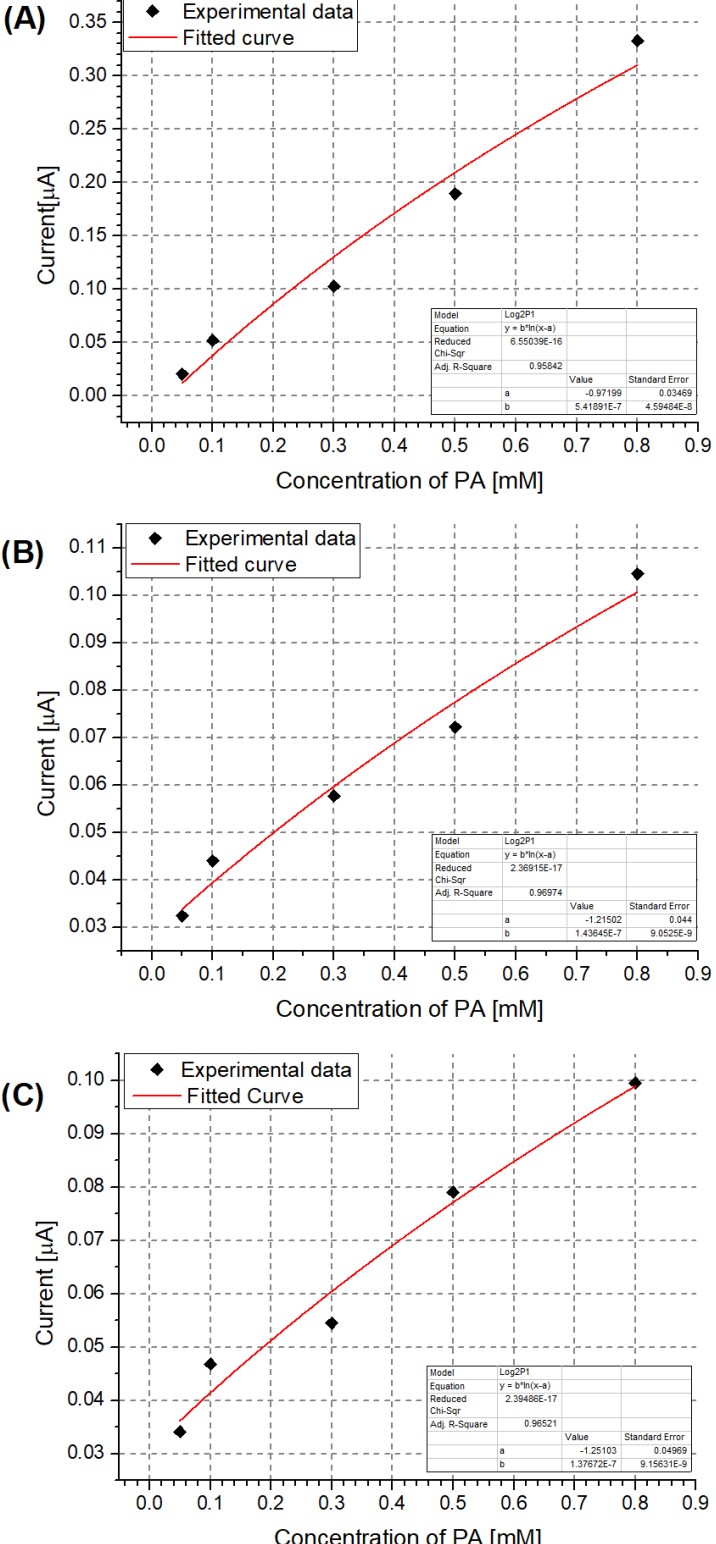

**Figure 6.** Calibration curves for the detection of PA using the investigated electrodes: (**A**) unmodified GC electrode; (**B**) NIP PCz/GC electrode; (**C**) MIP PCz/GC electrode.

### 3.3. Investigation of Electrode Cross-Selectivity

Investigation of cross-selectivity was conducted using two potential interferents: nitrobenzene and nitromethane, respectively deviating slightly and highly from the topology of picric acid (Figure 7). Cross-selectivity studies were conducted in DPV experiments analogous to the investigation of the response of the electrodes to PA, utilising higher concentrations of the interferents, so as to serve as benchmarks against the response of PA. The obtained results (Table 3) show that for the MIP layers deposited on Pt (Figure 8) and GC (Figure 9) electrodes, selectivity against nitromethane is high in both cases, but the high current signals observed for nitrobenzene indicate that for this interfering agent selectivity is limited.

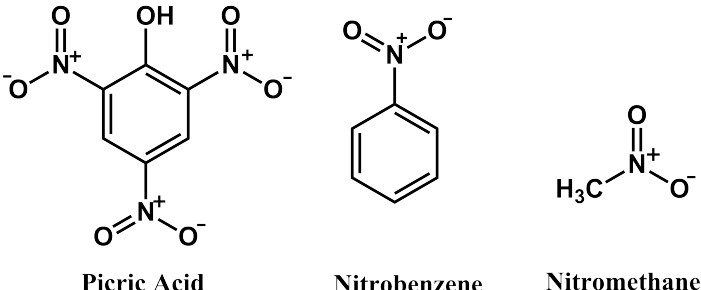

**Figure 7.** Schematic representation of the molecular structures of PA and selected interfering agents.

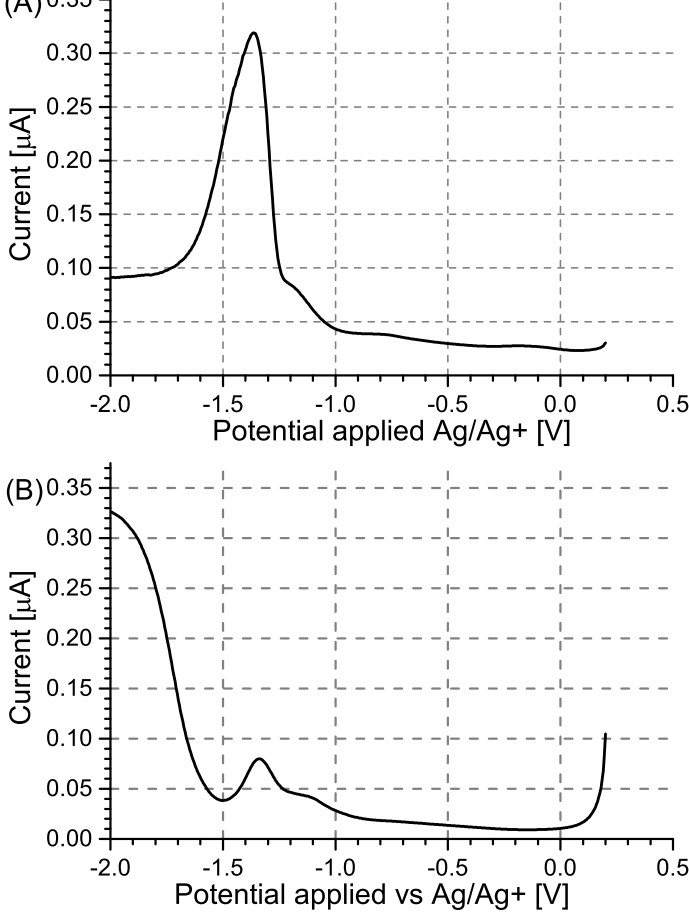

**Figure 8.** Differential pulse voltammograms recorded for MIP PCz/Pt electrodes. The voltammograms were recorded in $Bu_4NBF_4$/MeCN solutions containing (**A**) 9 mM nitrobenzene or (**B**) 18 mM nitromethane.

Interestingly, in the case of the NIP layer on Pt electrodes (Figure 10), a higher degree of selectivity against nitrobenzene is observed than in the case of the MIP layer, but lower selectivity to nitromethane is, in turn, observed. The fact that the NIP layer, whose pores are subject to a random size distribution, showed better selectivity to nitrobenzene than the MIP layer may stem from the fact that the pores of the MIP layer are tuned to the topology of PA. However, upon repeated doping and de-doping of the MIP, the shape of these pores deviates, becoming able to match both PA and the topologically similar nitrobenzene. This deviation, however, is insufficient to accommodate nitromethane, leading to the observed higher selectivity of the MIP than of the NIP against this interfering agent.

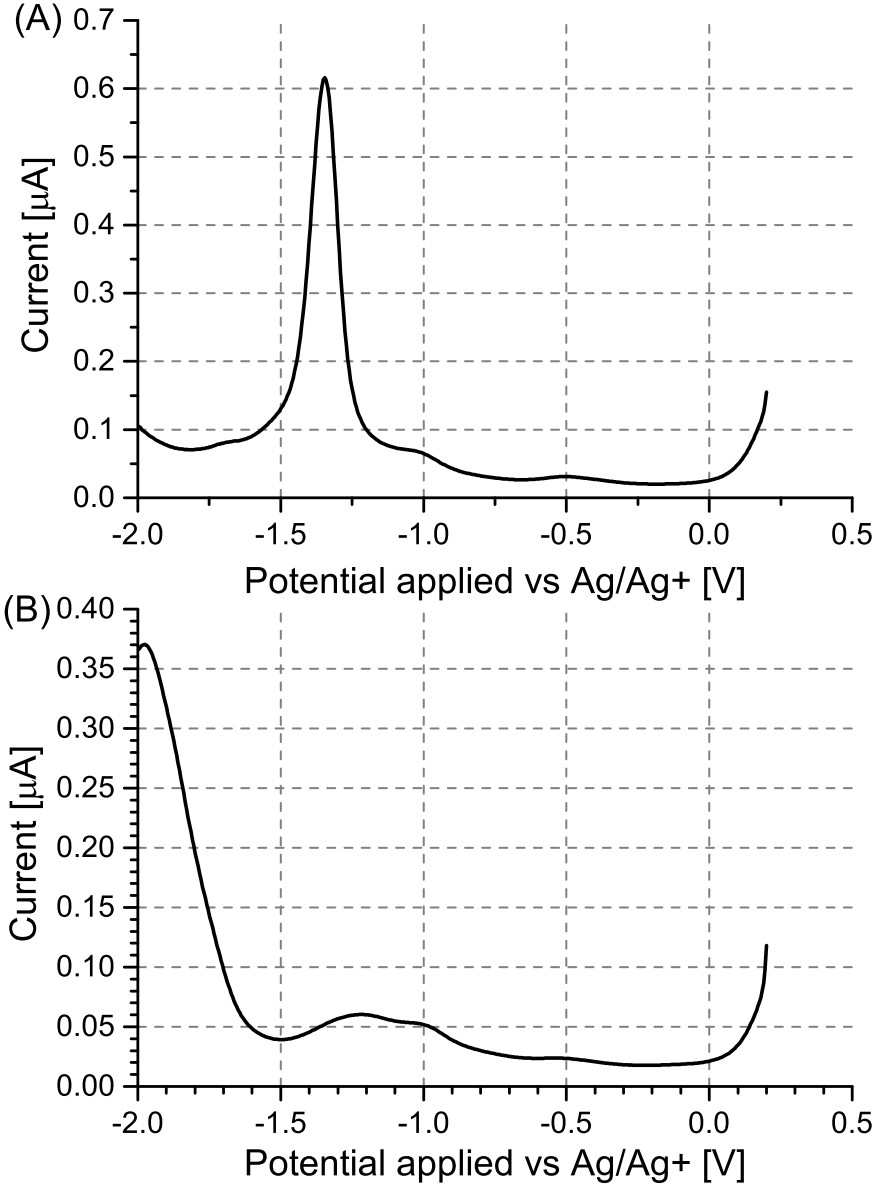

**Figure 9.** Differential pulse voltammograms recorded for MIP PCz/GC electrodes. The voltammograms were recorded in $Bu_4NBF_4$/MeCN solutions containing (**A**) 9 mM nitrobenzene or (**B**) 18 mM nitromethane.

**Table 3.** Comparison of reduction current values observed during DPV measurements of PA, nitrobenzene, and nitromethane reduction.

| Compound | NIP Cz/Pt | MIP Cz/Pt | IF Pt | MIP Cz/GC | IF GC |
|---|---|---|---|---|---|
| PA (0.8 mM) | 0.235 μA | 0.416 μA | 1.77 | 0.099 μA | 0.95 |
| Nitrobenzene (9 mM) | 0.056 μA | 0.319 μA | 5.70 | 0.616 μA | - |
| Nitromethane (18 mM) | 0.136 μA | 0.080 μA | 0.59 | 0.060 μA | - |

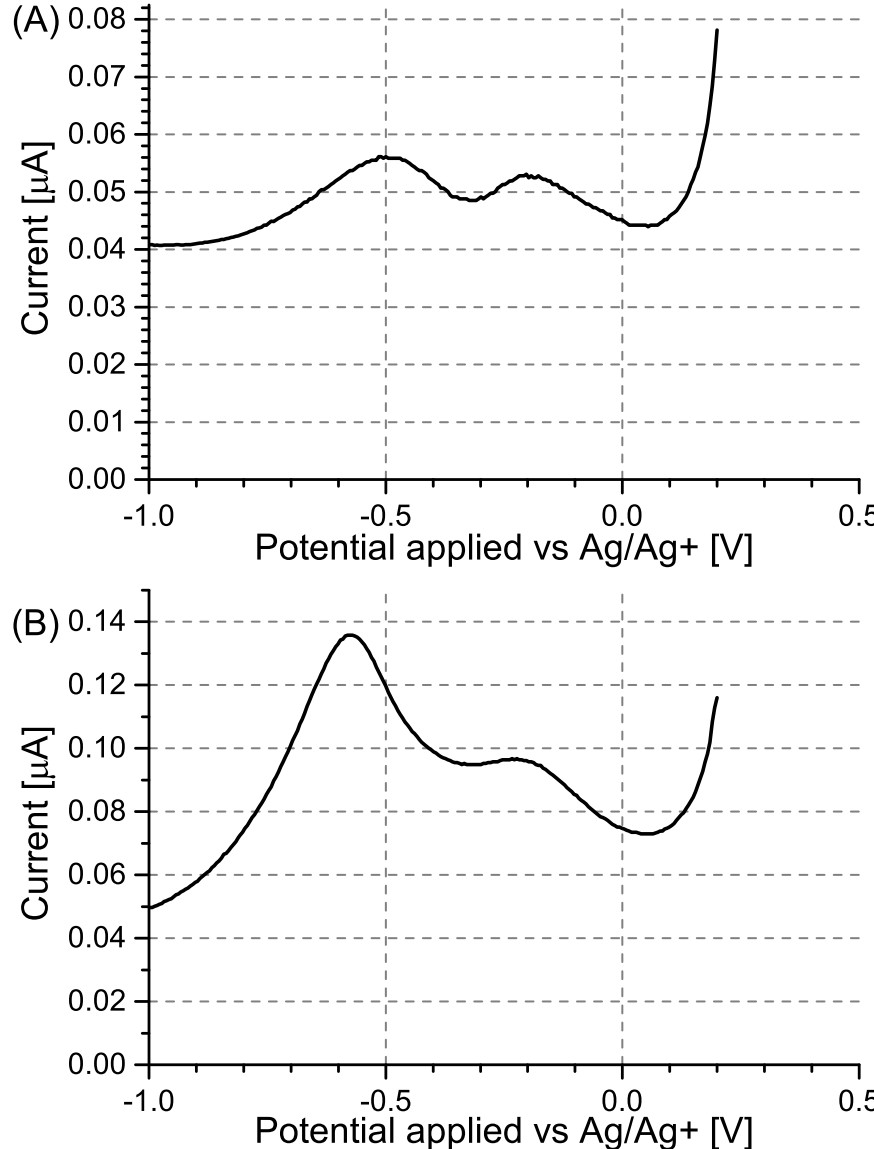

**Figure 10.** Differential pulse voltammograms recorded for NIP PCz/Pt electrodes. The voltammograms were recorded in $Bu_4NBF_4$/MeCN solutions containing (**A**) 9 mM nitrobenzene or (**B**) 18 mM nitromethane.

*3.4. Calculation of the Interactions between the Template and Monomer*

Quantum chemical calculations were performed so as to investigate the possible existence of interactions between the model structures of carbazole and PA. Three systems were optimised: (1) isolated carbazole, (2) isolated PA, and (3) carbazole in the presence of PA. The total energies of these systems were compared. We observed that the total energy of the combined system (Carbazole + PA) was lower than the sum of the energies of the isolated molecules (E3 < E1 + E2), indicating a stabilising interaction between carbazole

and picric acid. Geometry optimisation of the combined system (3) revealed the formation of a hydrogen bond between the oxygen atom in the nitro group of picric acid and the hydrogen atom attached to the nitrogen atom of carbazole, as illustrated in Figure 11.

The energy of this hydrogen bond was calculated to be 31.43 kJ/mol by using the following equation: $\Delta E = E3 - E1 - E2$. This result suggests a relatively strong interaction between these two molecules. It is worth noting that in the case of polycarbazole, one molecule of picric acid may form multiple hydrogen bonds with different carbazole repeat units.

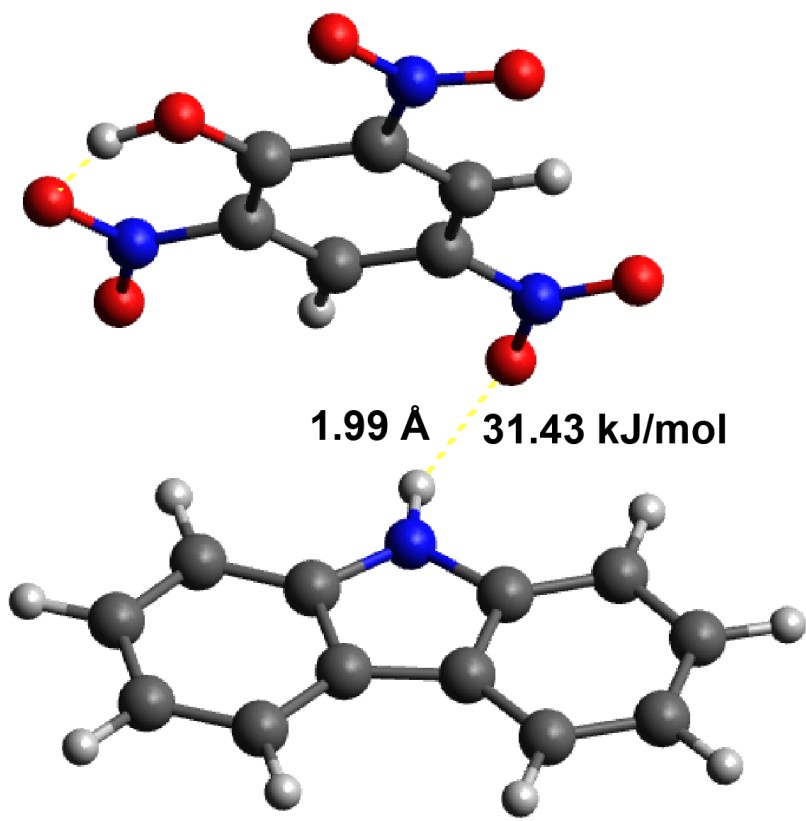

**Figure 11.** Calculated hydrogen bond between carbazole and PA using the B3LYP 3–11 G(d) function.

## 4. Conclusions

Quantum-mechanical calculations indicated that PA interacts strongly (31.43 kJ/mol) with both carbazole (monomer) and the repeat units of polycarbazole, which is typically sufficient for achieving a significant increase in the sensitivity of sensors due to molecular imprinting. Despite the existence of these interactions, the electrochemical detection results show only a marginal effect of molecular imprinting in the case of modified Pt electrodes (LODs of 0.26 and 0.62 mM, respectively, for MIP PCz/Pt and NIP PCz/Pt), translating into an imprinting factor of 1.77. Conversely, in the case of modified GC electrodes, molecular imprinting appeared to be counter-productive (IF = 0.95), as it results in an increase in the LOD values (0.57 and 0.12 mM, respectively, for MIP PCz/GC and NIP PCz/GC).

The very minor improvement of PA detection upon molecular imprinting likely stems from the fact that not only are the conjugated polymer chains highly rigid, but upon doping and de-doping, they undergo dearomatisation and rearomatisation, significantly changing their arrangement in space. This process likely leads to the gradual deformation of any pores remaining after the removal of the template, translating into a decrease in the performance of the MIP over time down to the NIP performance baseline.

The deformation of pores hypothesis is also supported by the results of cross-selectivity investigations, as the NIP layers show higher selectivity towards nitrobenzene than the MIP layers. The IF value calculated for the layers deposited on Pt and used to detect

nitrobenzene is 5.70, much higher than the value of 1.77 observed for PA. Conversely, the IF observed in the case of nitromethane is 0.59. These results indicate that while molecular imprinting increased the response of the layer towards nitroaromatics in general against nitroalkanes, it is not sufficiently selective to differentiate between nitrobenzene and PA. This feature can be attributed to the change in the shape of the pores present in the MIP layers, as contrasted to the random distribution of pore sizes in the NIP layers. Where the random distribution in the NIP layers allows pores of different sizes, imprinting increases the share of pores with sizes roughly corresponding to the size of the template molecule. Consequently, even though the pore shape begins deviating due to repeated doping and de-doping, pore size will remain roughly similar, explaining the observed IF values that were >1 for PA and nitrobenzene, as well as the IF < 1 value for nitromethane.

The lower performance of electrodes modified with either NIP or MIP PCz layers in comparison to that of the unmodified electrodes may be caused by the relatively lower conductivity of the conjugated polymer layers in comparison with either Pt or GC electrodes. Moreover, polycarbazole typically produces layers that vary significantly in thickness, due to its nucleation mode, which may also hinder the adsorption of the planar and highly polar PA molecules on the surface of this polymer in comparison with the highly planar PT and GC electrode surfaces.

Taking the above into consideration, two main factors necessary for the successful use of molecularly imprinted conjugated polymers can be postulated. Firstly, during electrochemical polymerisation, the precipitating polymer film must not undergo repeated doping/de-doping, as this process appears to distort the size and shape of the existing pores, as discussed above. This is evidenced by the fact that molecularly imprinted polycarbazole derivatives were utilised as receptor layers for sensors when their electrodeposition did not involve their de-doping [41]. This factor can also explain the very broad application of polypyrrole-based MIP sensors, as polypyrroles undergo de-doping only at very strongly negative potentials, usually exhibiting a similar doping state across the typical conditions of their electrosynthesis process. Secondly, a conjugated polymer with a nucleation mode more suited to the template molecule should be used so as to promote the adsorption of the template onto the surface of the molecularly imprinted conjugated polymer film.

**Author Contributions:** Conceptualization: K.G. and T.J.; Methodology: K.G.; Investigation: K.G., M.F., P.J., W.K., A.S. and T.J.; Validation: A.S. and T.J.; Formal analysis: K.G. and T.J.; Data curation: K.G.; Visualization: M.F., W.K. and A.S.; Writing—original draft preparation: K.G., M.F., A.S. and T.J.; Writing—review and editing: K.G., A.S. and T.J.; Supervision: K.G. All authors have read and agreed to the published version of the manuscript.

**Funding:** T.J. acknowledges the grant for starting research on a new subject (no. 04/040/SDU/10-22-06) from the Rector of the Silesian University of Technology. T.J. acknowledges the scientific and innovative merit grant no. 04/040/RGJ24/0278 of the Rector of Silesian University of Technology. A.S. acknowledges the scientific and innovative merit grant no. 04/040/RGJ24/0275 of the Rector of Silesian University of Technology.

**Institutional Review Board Statement:** Not applicable.

**Informed Consent Statement:** Not applicable.

**Data Availability Statement:** Data are available from the authors upon request.

**Conflicts of Interest:** The authors declare no conflicts of interest.

## Appendix A

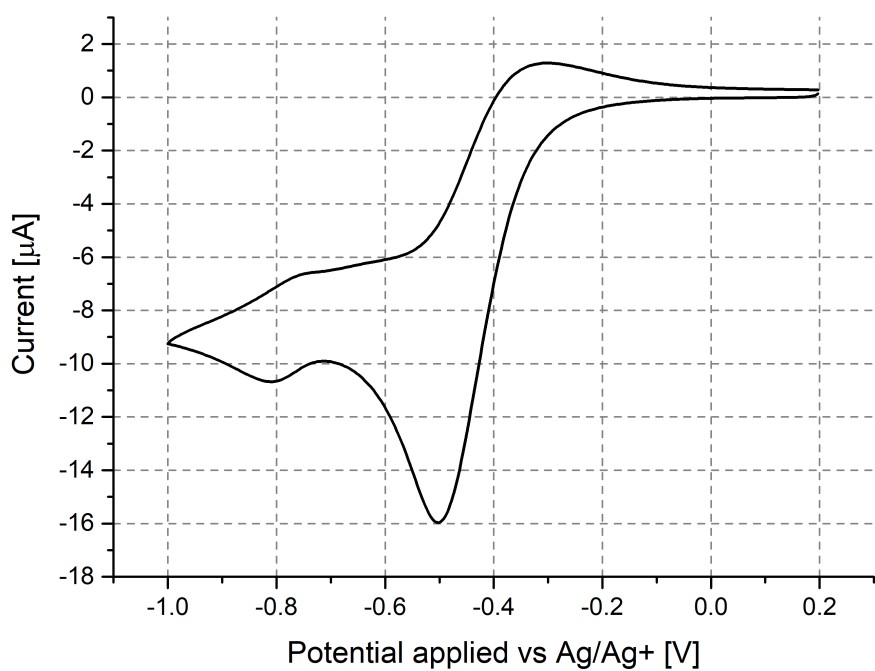

**Figure A1.** Cyclic woltammetry of 5 mM PA in 0.1 M Bu$_4$NBF$_4$/MeCN. The voltammogram was recorded at a potential range of 0.2 to −1 V, and the potential scan rate was 0.1 V/s.

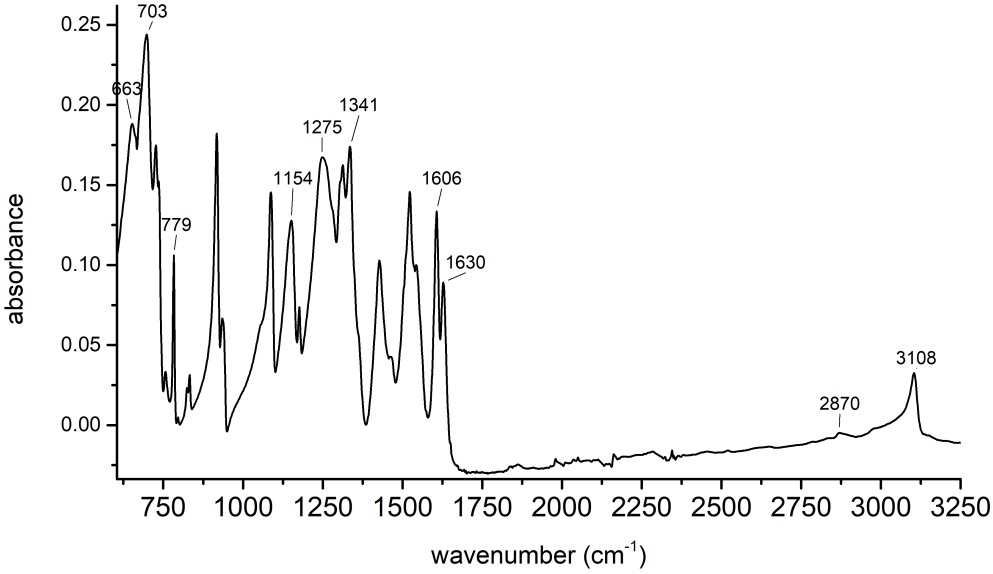

**Figure A2.** IR-ATR spectrum of PA.

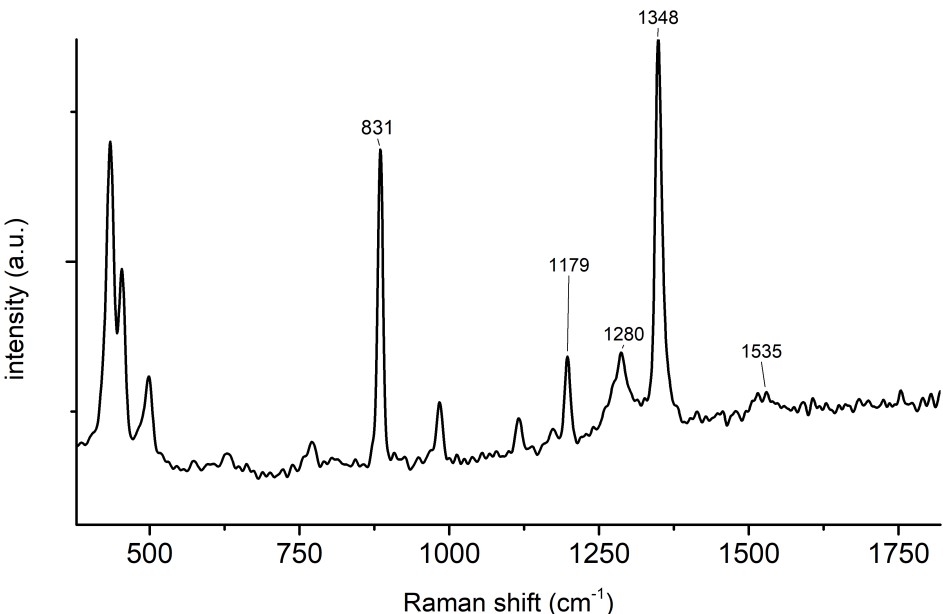

**Figure A3.** Raman spectrum of PA.

**Table A1.** Optimised coordinates (XYZ in Angstroms) for carbazole.

| | Carbazole | | |
|---|---|---|---|
| N | 0.02981 | 1.82223 | −0.00815 |
| C | −1.11644 | 1.02826 | −0.00910 |
| C | −0.71386 | −0.34060 | −0.01061 |
| C | 0.74168 | −0.35128 | −0.01054 |
| C | 1.16430 | 1.01151 | −0.00900 |
| C | 2.51851 | 1.35112 | −0.00852 |
| C | 3.45290 | 0.31688 | −0.00961 |
| C | 3.05074 | −1.03126 | −0.01113 |
| C | 1.69857 | −1.37045 | −0.01161 |
| C | −1.68551 | −1.34569 | −0.01177 |
| C | −3.03260 | −0.98678 | −0.01142 |
| C | −3.41495 | 0.36711 | −0.00993 |
| C | −2.46558 | 1.38760 | −0.00875 |
| H | −2.76961 | 2.42780 | −0.00760 |
| H | −4.46861 | 0.62305 | −0.00968 |
| H | −3.79597 | −1.75596 | −0.01231 |
| H | −1.39387 | −2.39031 | −0.01294 |
| H | 4.51023 | 0.55737 | −0.00926 |
| H | 2.83764 | 2.38678 | −0.00735 |
| H | 1.39172 | −2.41065 | −0.01280 |
| H | 3.80272 | −1.81159 | −0.01195 |
| H | 0.03719 | 2.83358 | −0.00696 |

**Table A2.** Optimised coordinates (XYZ in Angstroms) for PA.

| Picric Acid | | | |
|---|---|---|---|
| C | 0.00945 | 3.33411 | 0.03651 |
| C | −1.17395 | 2.54845 | 0.02991 |
| C | −1.12209 | 1.17076 | 0.02080 |
| C | 0.10007 | 0.51084 | 0.01801 |
| C | 1.28347 | 1.21765 | 0.02437 |
| C | 1.23781 | 2.60246 | 0.03349 |
| N | 2.49980 | 3.30598 | 0.04006 |
| N | −2.51177 | 3.15454 | 0.03243 |
| N | 0.13368 | −0.94626 | 0.00818 |
| O | −2.59304 | 4.42698 | 0.04244 |
| O | −3.50351 | 2.33420 | 0.02650 |
| O | −0.99687 | −1.54952 | 0.00292 |
| O | 2.47491 | 4.62626 | 0.04753 |
| H | −2.04592 | 0.61352 | 0.01596 |
| H | 2.23067 | 0.70114 | 0.02231 |
| O | −0.01358 | 4.66298 | 0.04530 |
| H | 0.98294 | 4.97739 | 0.04855 |
| O | 3.57888 | 2.63791 | 0.03726 |
| O | 1.29258 | −1.49580 | 0.00624 |

**Table A3.** Optimised coordinates (XYZ in Angstroms) for the investigated PA–carbazole complex.

| Carbazole + Picric Acid | | | |
|---|---|---|---|
| C | 0.78977 | 3.54467 | 2.65822 |
| C | −0.62494 | 3.42763 | 2.71890 |
| C | −1.39194 | 3.36173 | 1.57606 |
| C | −0.79390 | 3.39592 | 0.32093 |
| C | 0.57713 | 3.48852 | 0.20030 |
| C | 1.35118 | 3.57474 | 1.34362 |
| N | 2.77608 | 3.71222 | 1.16349 |
| N | −1.34928 | 3.37843 | 3.99743 |
| N | −1.61719 | 3.36607 | −0.86974 |
| O | −0.66607 | 3.42812 | 5.07247 |
| O | −2.62944 | 3.28279 | 3.91711 |
| O | −2.87847 | 3.25082 | −0.71841 |
| O | 3.53374 | 3.77284 | 2.23971 |
| H | −2.46434 | 3.29022 | 1.67508 |
| H | 1.05438 | 3.49186 | −0.76596 |
| O | 1.54920 | 3.62212 | 3.74411 |
| H | 2.53593 | 3.70336 | 3.41606 |
| O | 3.25032 | 3.76931 | −0.01722 |
| O | −1.01095 | 3.48576 | −2.01032 |
| C | 0.77076 | 7.08961 | −1.22926 |
| N | −0.20516 | 6.21119 | −1.69641 |
| C | −1.41935 | 6.88531 | −1.82469 |
| C | −1.22820 | 8.24325 | −1.43832 |
| C | 0.17149 | 8.37596 | −1.05892 |
| C | 0.94539 | 9.44130 | −0.58998 |
| C | 2.29053 | 9.22493 | −0.29392 |

**Table A3.** *Cont.*

| Carbazole + Picric Acid | | | |
|---|---|---|---|
| C | 2.86441 | 7.95078 | −0.45292 |
| C | 2.11506 | 6.86993 | −0.91808 |
| C | −2.66552 | 6.40365 | −2.23297 |
| C | −3.72888 | 7.30395 | −2.26060 |
| C | −3.55676 | 8.65113 | −1.88917 |
| C | −2.31301 | 9.12582 | −1.47445 |
| H | −2.18901 | 10.16273 | −1.18224 |
| H | −4.40587 | 9.32389 | −1.92215 |
| H | −4.70834 | 6.95814 | −2.57062 |
| H | −2.79492 | 5.36025 | −2.49568 |
| H | 2.90159 | 10.04348 | 0.06773 |
| H | 0.50375 | 10.42276 | −0.45749 |
| H | 2.56567 | 5.88758 | −1.00781 |
| H | 3.90976 | 7.80228 | −0.20651 |
| H | −0.08465 | 5.24726 | −1.98864 |

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
