# Peer review of "The Failure of Molecular Imprinting in Conducting Polymers: A Case Study of Imprinting Picric Acid on Polycarbazole"

_sensors, doi:10.3390/s24020424_

Round 1

Reviewer 1 Report

Comments and Suggestions for Authors

The article by Tomasz Jarosz and co-authors is devoted to the study of the picric acid detection in solution by polycarbazolole-modified and non-modified electrodes.

However, before the article can become suitable for publication, several serious issues must be resolved:

 1. What was the point of synthesizing picric acid if it is available as a commercial product? Moreover, if the authors want to provide proof of the structure of the resulting product, then the 1H NMR spectrum alone is not enough. It is necessary to at least determine the melting point and provide elemental analysis data.

2. Figures 5, 7, 8, and 9 are not condemned in any way in the article. They are not even mentioned in the text.

3.  Please check the total energy values in Table 3. They are too high.

4. Unfinished sentence in the conclusion section: This is random distribution...

5. In the section discussing the cross-selectivity of nitrobenzene electrodes, the authors should make a comparison with literature data: https://doi.org/10.1016/j.foodchem.2021.131279. Where the reasons for high sensitivity to nitrobenzene were discussed based on scanning electron microscopy data and studying the surface morphology of screen-printed electrode modified with polycarbazole.

It would also be interesting if the authors managed to take similar photographs of their electrodes and understand their morphology.

Comments on the Quality of English Language

Minor editing of English language required.

Reviewer 2 Report

Comments and Suggestions for Authors

The author prepared molecularly imprinted polycarbazole layers for the detection of picric acid (PA). They investigated the process for producing a MIP polycarbazole on platinum and glassy carbon electrodes and investigated their performance in detecting PA. The limit of detection was found to be 210 0.26 and 0.62 mM respectively for MIP PCz/Pt and NIP PCz/Pt. Conversely, they reported that in the case of modified GC electrodes, molecular imprinting appeared to be counter-productive, as it resulted in increasing the LOD value (0.57 and 0.12 mM respectively for MIP PCz/GC and 213 NIP PCz/GC). The manuscript is well organized. The presented paper is interesting, but the following corrections should be done before publishing.

Below are my concerns and suggestions to improve the manuscript;

The abstract is missing a piece of brief information on effects of picric acid.

The abstract needs to be highly quantitative.

The introduction is missing a piece of brief information on the permissible limits of PA in the matrix. Please provide more details about the PA effects in the introduction. 

I recommend adding the following current references related to the MIP in the introduction of this manuscript to improve its update,

https://doi.org/10.3390/chemosensors11060318; https://doi.org/10.3390/polym15030629; https://doi.org/10.1016/j.jpba.2022.115213; https://doi.org/10.3390/chemosensors11070380

The recognition mechanism should be discussed in more detail. 

Please, in detail explain the chemical stability of the MIPs.

In Figure 4, the blank solution line should be included in the graph.

Reproducibility studies are very important for this device. The authors must explain the advantages. 

The imprinting factor (IF) is calculated by comparing the recognition of template analyte on the imprinted polymer with a comparable non-imprinted polymer. IF must be added in manuscript.

It is best to describe the performance and benefits of the proposed sensor in detail with other studies in the text.

The conclusion needs to be highly quantitative and should be discussed in more detail. Please improve the conclusion with clear quantitative findings.

Round 2

Reviewer 1 Report

Comments and Suggestions for Authors

The authors have made an efficient revision on their work, which can be accepted in the current form.

Comments on the Quality of English Language

Minor editing of English language required.

Author Response

Esteemed Sir / Madam, (Reviewer 1)

We would like to thank you for your perusal of our manuscript and your comments, which have helped us improve its quality and clarity.

Most cordially

Reviewer 2 Report

Comments and Suggestions for Authors

Authors have carefully checked and modified this manuscript. Now it can be accepted for publication in this journal without further revision.

Author Response

Esteemed Sir / Madam, (Reviewer 2)

We would like to thank you for your perusal of our manuscript and your comments, which have helped us improve its quality and clarity.

Most cordially